# Properties and Crystal Structure of the *Cereibacter sphaeroides* Photosynthetic Reaction Center with Double Amino Acid Substitution I(L177)H + F(M197)H

**DOI:** 10.3390/membranes13020157

**Published:** 2023-01-26

**Authors:** Tatiana Yu. Fufina, Georgii K. Selikhanov, Azat G. Gabdulkhakov, Lyudmila G. Vasilieva

**Affiliations:** 1Federal Research Center Pushchino Scientific Center for Biological Research PSCBR, Institute of Basic Biological Problems, Russian Academy of Sciences, Institutskaya Street 2, 142290 Pushchino, Russia; 2Institute of Protein Research, Russian Academy of Sciences, Institutskaya Street 4, 142290 Pushchino, Russia

**Keywords:** purple bacteria, membrane pigment-protein complex, photosynthetic reaction center, site-directed mutagenesis, crystal structure

## Abstract

The photosynthetic reaction center of the purple bacterium *Cereibacter sphaeroides* with two site-directed mutations Ile-L177–His and M197 Phe–His is of double interest. The substitution I(L177)H results in strong binding of a bacteriochlorophyll molecule with L-subunit. The second mutation F(M197)H introduces a new H-bond between the C2-acetyl carbonyl group of the bacteriochlorophyll P_B_ and His-M197, which is known to enhance the stability of the complex. Due to this H-bond, π -electron system of P finds itself connected to an extensive H-bonding network on the periplasmic surface of the complex. The crystal structure of the double mutant reaction center obtained with 2.6 Å resolution allows clarifying consequences of the Ile L177 – His substitution. The value of the P/P^+^ midpoint potential in the double mutant RC was found to be ~20 mV less than the sum of potentials measured in the two RCs with single mutations I(L177)H and F(M197)H. The protein environment of the BChls P_A_ and B_B_ were found to be similar to that in the RC with single substitution I(L177)H, whereas an altered pattern of the H-bonding networks was found in the vicinity of bacteriochlorophyll P_B_. The data obtained are consistent with our previous assumption on a correlation between the bulk of the H-bonding network connected with the π-electron system of the primary electron donor P and the value of its oxidation potential.

## 1. Introduction

Photosynthetic reaction centers are integral membrane complexes responsible for the extremely efficient conversion of light energy into the chemical energy of separated charges. The crystal structure of the reaction center (RC) of the purple nonsulfur bacterium *Cereibacter* (*C.*) *sphaeroides* shows that this complex consists of three subunits and ten cofactors (Figure 1A) [1]. The L and M protein subunits and cofactors are related to each other by an approximate two-fold symmetry axis [2]. Cofactors are represented by four bacteriochlorophylls (BChl), two bacteriopheophytins (BPeo), two quinones, one carotenoid molecule, and a non-heme iron atom and are arranged in two membrane-spanning branches, A and B. Only one, A branch, is active in photosynthetic electron transfer. Two BChls, P_A_ and P_B_, are combined into a dimer P that serves as a primary electron donor. An important property of P is the midpoint potential (E_m_) P/P^+^ which is related to the energy level of P* after light excitation. In the wild-type RCs of purple bacteria, cofactors interact with integral L- and M-subunits through relatively weak non-covalent contacts. Figure 1B shows four BChl molecules in the *C. sphaeroides* RC and some amino acid residues from their environment. Histidine residues, L173 and M202, coordinate central Mg^2+^ atoms of BChls P_A_ and P_B_, respectively, and His-M182 is the ligand of BChl B_B_. In native *C. sphaeroides* RC, the C2-acetyl carbonyl group of the BChl P_A_ forms an H-bond with His-L168. The C2-acetyl carbonyl group of BChl P_B_ is free of H-bonds with its environment since the residue closest to this group and symmetry related to His-L168 is Phe-M197 [3]. Crystal structures of photosynthetic RCs also contain a considerable amount of bound water molecules with a few of them located in the hydrophobic core of the complexes [4] (not shown, discussed below).

Previously, it was shown that substitution of Ile-L177 for His in the RC of *C. sphaeroides* near BChl B_B_ and P_A_ resulted in unexpectedly strong binding between one of these chromophores and L-subunit, which was stable during SDS-PAGE under a variety of denaturing conditions [5]. The appearance of a new band at 638 nm in the absorption spectrum of the I(L177)H mutant RC indicated that the central Mg^2+^ atom of one out of four BChls became hexa-coordinated, while the other BChls remained penta-coordinated [6]. Treatment of the isolated RC with sodium borohydride revealed that the pigment strongly bound to the protein is BChl P_A_ [7]. Later, the crystal structure of RC I(L177)H was obtained with a relatively low resolution of 2.9 Å [6]. The difference electron density map showed low sigma peaks adjacent to the central Mg^2+^ atom of BChl B_B_, indicating the potential presence of a water molecule. Computer modeling proposed the involvement of two water molecules in the β-coordination of the BChl B_B_, however, a higher resolution structure is needed for a conclusion on this issue. The question concerning the nature of the strong attachment of BChl P_A_ to the RC protein also remains open.

We suggested that decreased thermal stability of the I(L177)H mutant RC structure, shown in previous work [7], could be one of the possible reasons for the poor quality of the RC crystals and the resulting insufficient resolution of the crystal structure [6]. It is known that H-bond formation between the protein environment and the C2-acetyl carbonyl groups of the BChl dimer contributes to the strengthening of the local structure of the *C. sphaeroides* RC, increasing its resistance to elevated temperatures and pressure [8,9]. Based on the recently resolved crystal structure of the RC F(M197)H, it was found that the increased stability of this complex is associated with the incorporation of the C2-acetyl carbonyl group of BChl P_B_ into the H-bonding network surrounding this cofactor. It was also suggested that the size and pattern of the H-bonding networks near P can affect the oxidation potential of the BChl dimer [10]. However, more experimental data are needed to support this assumption. While the H-bonding networks associated with proton transfer and secondary quinone reduction are well characterized, the H-bonding clusters on the periplasmic side of the purple bacteria RCs are less studied and their role requires further investigation.

We have combined I(L177)H and F(M197)H mutations in *C. sphaeroides* RC to increase the stability of its structure and thereby increase the probability of obtaining the crystal structure of this complex with higher resolution. This work introduces the new double mutant RC and describes its spectral properties and oxidation potential of the primary electron donor P. X-ray crystal structure of the RC I(L177)H + F(M197)H, obtained with a resolution of 2.6 Å, clarifies some details of unusually strong pigment-protein interaction resulting from the Ile→His substitution at L177 position and shows that a new water molecule appears from β–side of BChl B_B_, which is responsible for hexa-coordination of the central Mg^2+^ atom of this cofactor. In addition, the crystal structure provides data in support of the previously stated assumption about the correlation between the bulk of the H-bonding network in the vicinity of the BChl dimer and the value of the oxidation potential of the primary electron donor.

## 2. Experimental

### 2.1. Construction of the Mutant Strain

The I(L177)H and F(M197)H mutations previously introduced [11,12] were combined together in *puf*-operon by simple cloning using available unique restriction sites [11]. Poly-histidine tag (6His) was introduced from the 3′- end of the *puf*-M gene, similar to what was previously described [13]. The modified *puf*-operon with the mutations was cloned into the broad-host-range vector pRK415, containing DNA fragment that included the *puf*LMX genes. The resulting plasmid was transferred into *C. sphaeroides* strain DD13 [14] through conjugative crossing to give a recombinant strain with the RC-only phenotype. The control pseudo wild-type strain was similarly made using the DD13 strain, complemented by a wild-type copy of the *puf*LMX genes.

### 2.2. Bacterial Growth and Reaction Center Purification

The growth of recombinant bacterial strains under dark semi-aerobic conditions has been described elsewhere [11]. The cells were harvested and then broken by ultrasonication, and membranes were sedimented by ultracentrifugation. Reaction centers were solubilized using lauryldimethylamine oxide (LDAO) (Sigma-Aldrich, Missouri, USA) and then purified by passage through Ni-NTA agarose column (Serva, Heidelberg, Germany). After the purification of the RCs, the detergent was replaced as previously described [15]. The purified RCs were dissolved in 20 mM Tris-HCl buffer (pH 8.0) (Helicon, Moscow, Russia) containing 0.2% sodium cholate (Sigma-Aldrich, Missouri, USA) or 0.05 % Triton X-100 (Helicon, Moscow, Russia).

### 2.3. Thermal Stability, Adsorption Spectroscopy, and Redox Potential Measurements

Thermal stability of the RCs was studied in Tris-Triton buffer with 250 mM NaCl by monitoring thermo-dependent changes of the Q_Y_ band amplitude at 48 °C as previously described [15]. Absorption spectra were recorded using a Shimadzu UV-1800 spectrophotometer (Shimadzu, Kyoto, Japan) at room temperature. Sodium ascorbate (1 mM) was added to samples to maintain the primary electron donor in a reduced state. The P/P^+^ midpoint potential was determined by chemical titrations using potassium ferricyanide and sodium ascorbate as previously described [6].

### 2.4. Reaction Center Crystallization and Data Analysis

Trigonal crystals of the reaction center with the double mutation I(L177)H + F(M197)H, space group P3_1_21, were grown using the hanging drop vapor diffusion method as previously described in detail [6]. X-ray diffraction data were collected at the P11 beamline PETRA III electron storage ring (Hamburg, Germany) using cryo-cooled crystals. The crystal used for data collection diffracted to a higher resolution limit of 2.6 Å. The diffraction data were processed and scaled using XDS [16]. Molecular replacement was performed with Phaser [17], using the coordinates of the I(L177)H mutant RC structure from *C. sphaeroides* (PDB ID: 3V3Z) [6] as the search model. Data collection and refinement statistics are given in Table 1. Rigid body refinement was carried out before refinement using restrained maximum likelihood refinement in REFMAC 5.0 [18]. Manual rebuilding of the model was carried out in Coot [19]. The final cycle with an occupancy refinement was performed in Phenix [20]. The coordinates and structure factors of double mutant RC have been deposited in the Protein Data Bank (PDB ID: 8C3F). Figures were prepared using the program PyMol [21].

## 3. Results

### 3.1. Absorption Spectra

The mutant RC with the double substitution I(L177)H + F(M197)H was stably assembled in photosynthetic membranes and the yield of the RC after purification was similar to that of the wild-type RC from the same amount of the cell mass. The absorption spectrum of the wild-type RC measured at room temperature shows the Q_Y_ P band at 865 nm, the Q_Y_ B band at 804 nm, and the Q_Y_ H band at 759 nm [22]. In the short wavelength region, the BChls and BPheos of the wild-type RC give rise to asymmetric absorbance bands with main peaks at 599 and 533 nm, respectively. Previously, it was shown that the absorption spectrum of RC F(M197)H was similar to the spectrum of the wild-type RC [23]. As described by [11], the room temperature absorption spectrum of RC I(L177)H showed a noticeable 18 nm blue shift and a diminished amplitude of the Q_Y_ P band near 847 nm, a diminished amplitude of the Q_Y_ B band, as well as decreased absorption in the Q_X_ region of BChl absorption near 600 nm. The absorption spectrum of RC I(L177)H + F(M197)H showed similarity to that of RC I(L177)H, except that the blue shift of the Q_Y_ P band was less pronounced and the maximum of this band was observed at 854 nm (Figure 2).

### 3.2. Thermal Stability

Thermal stability of the wild-type RC as well as mutant RCs I(L177)H, F(M197)H, and I(L177)H+F(M197)H was determined by recording thermo-dependent changes of the Q_Y_ B band amplitude at 48 °C in Tris-Triton buffer. It was shown that after 60 min of incubation in the spectrum of the wild-type RC, the amplitude of the Q_Y_ B band at 804 nm decreased by 50%, whereas the spectra of the RCs I(L177)H, F(M197)H, and I(L177)H+F(M197)H demonstrated 55%, 40%, and 45% decreases, respectively (Figure 3). These data show that, as expected, the addition of the F(M197)H mutation increases the thermal stability of RC I(L177)H, however, it remains lower than that of the single mutant RC F(M197)H.

### 3.3. P/P^+^ Midpoint Potential

The value of the P/P^+^ midpoint potential (E_m_ P/P^+^) of the double mutant RC was determined by redox titrations as 555 ± 10 mV (Appendix A). Previously, it was shown that the single mutation F(M197)H raises the value of E_m_ P/P^+^ by 125 mV [24] due to the formation of an H-bond between the C2-acetyl carbonyl group of BChl P_B_ and the imidazole group of His-M197, whereas the substitution I(L177)H results in a decrease of the E_m_ P/P^+^ value by 50 mV [6]. Assuming that in the double mutant RC each of the mutations causes the same structural changes as in each RC with single substitutions, one would expect the resulting E_m_ P/P^+^ value to be ~575 mV. The crystal structure of RC I(L177)H + F(M197)H resolved in this study allows us to suggest a possible reason for the discrepancy between expected and experimentally observed mid-point potential P/P^+^ (discussed below).

### 3.4. Crystal Structure

The crystal structure of the double mutant reaction center was obtained with a resolution of 2.6 Å and an estimated maximal coordinate error of 0.31 Å (Table 1). After refinement, the structural model of RC I(L177)H + F(M197)H was compared with the structures of the wild-type RC (PDB ID: 3V3Y), RC F(M197)H (PDB ID: 7OD5), and I(L177)H (PDB ID: 3V3Z). In the structure of RC I(L177)H + F(M197)H, the protein environment of BChls B_B_ and P_A_ is similar to that in the previously described structure of RC I(L177)H [3]. A new electron density appears from the β-side of B_B_, which suggests the incorporation of a new water molecule (Figure 4A). The structure shows that the water is located at a distance of 2.7 Å from the central Mg^2+^ atom of B_B_, which is typical for the BChl axial ligands. This water is also 3.1 Å close to the imidazole group of His-L177, which is within the typical range for H-bond formation (Figure 4).

The imidazole group of His-L177 points towards the methyl carbon of the pyrrole ring I of P_A_ with the center-to-center distance between N and C atoms of 2.06 Å. According to the electron density map, imidazole forms a covalent bond with the methyl carbon of P_A_ with a high degree of probability. Based on the structural data, we modeled pigment-protein interactions that occur as a result of Ile→His substitution at the L177 site (Figure 4B).

Comparison of the structures of the double mutant RC and RC F(M197)H in the proximity of BChl P_B_ shows structural conservation in the main body of the protein as well as in the cofactor system. In both complexes, the imidazole group of His-M197 is located within the H-bond distance from the C2-acetyl carbonyl group of BChl P_B_. It was noted that the C2-acetyl group shows 29.7° out of plane rotation (Figure 5), which is more significant compared with that observed in RC F(M197)H (20.7°) [10]. The data are consistent with spectral properties of RC I(L177)H + F(M197)H (as discussed below).

Analysis of the RC I(L177)H + F(M197)H structure in the vicinity of P_B_ (Figure 6) shows that the main pattern of the previously described H-bonding network, which includes the C2-acetyl carbonyl group of BChl P_B_, the imidazole group of His-M197, side groups of Asn-M195, Ser-L158, Asp-L155, and Tyr-M198 and two water molecules (assigned as waters-C and - E in Figure 6 and [10]), is similar to that observed in the RC with the single substitution F(M197)H. The sigma level of the electron density map peak for the water-C is low (1.7) and not sufficient to assert its presence with confidence. However, this water is observed in the structures of the wild-type RC and RC F(M197)H [10] as part of their H-bonding networks. As shown in Figure 6, there is a high degree of probability that water-C is also present in the structure of the double mutant RC. The distances between the related side groups in the network are all within the range of 2.5– 3.03 Å consistent with H-bond formation (Figure 6). The only exception is the distance between water-C and the side group of Asp-L155, which exceeds 3.7 Å. Taking into account the coordinate error, it suggests that no H-bond forms between these groups in RC I(L177)H + F(M197)H. The absence of this H-bond is consistent with some shortening of the distance between water-C and the imidazole group of His-M197 (2.7 Å), compared with the structure of RC F(M197)H (3.1 Å) (Figure 6). In addition, Figure 6B shows a change in the mutual arrangement of the Tyr-M198 and Asp-L155 side groups, indicating that Asp-L155 is shifted away from water-C in the double mutant RC structure.

It has been shown that in the wild-type RC of *C. sphaeroides,* an H-bonding network also presents on the periplasmic side of BChl P_A_ [10]. The network includes the C2-acetyl carbonyl group of BChl P_A_, the imidazole group of His-L168, side groups of Asn-L166, Asp-M184, and Tyr-L169 and a water molecule assigned as water-D. In the crystal structure of RC I(L177)H + F(M197)H, the positions of the side groups of the aforementioned residues did not change significantly, compared with those in the wild-type and F(M197)H RC structures (Appendix A). Although the sigma level of the electron density map peak for the water-D is low (1.5), its presence is indicated by the distance of this water from the side groups of Asn-L166 and Asp-M184, which is within the range of H-bond distances, similar to those in the wild-type and F(M197)H RCs. Thus, the data obtained show that the protein environment of BChls P_A_ and B_B_ is similar to that in the RC with single substitution I(L177)H.

## 4. Discussion

We have characterized spectral properties and mid-point potential P/P^+^ of the double mutant RC I(L177)H + F(M197)H as well as presented the crystal structure of the complex with 2.6 Å resolution. It is shown that the spectral properties of the double mutant RC are similar to those of RC I(L177)H, except that the blue shift of the Q_Y_ P band is less pronounced, namely 11 nm compared with 18 nm for RC I(L177)H. It was noticed that while a single mutation F(M197)H did not lead to evident changes in the spectral properties of RC, the addition of this mutation to RC I(L177)H led to a significant red shift of the Q_Y_ P band. This spectral shift of the Q_Y_ P band is likely a result of a much more substantial out-of-plane rotation of the C2-acetyl carbonyl group of BChl P_B_ (29.7º) compared with that observed in the structure of RC F(M197)H (20.7º) under the same conditions. As discussed previously in detail [10], the energy of the P Q_Y_ transition in the mutant RC F(M197)H appears to be affected both by the rotation of the acetyl group of P_B_, which causes a red shift of the P Q_Y_ band, and by a small increase in the distance between the two BChls of the dimer that leads to a blue shift of the band. In RC F(M197)H, these opposing effects compensate for each other, and as a result, the absorption spectrum remained practically the same as in the wild-type RC [23]. The resolution of the double mutant RC crystal structure does not allow us to estimate possible changes in the distance between the BChl planes in the dimer P but we see much more pronounced out-of-plane rotation of the acetyl group of BChl P_B_ and suggest that it causes an additional 7 nm red-shift of the P Q_Y_ band compared to the absorption spectrum of RC I(L177)H.

The original goal of this work was to combine RC I(L177)H and F(M197)H mutations to increase the stability of the mutant RC to improve the resolution of its crystal structure. The data obtained demonstrate that RC I(L177)H + F(M197)H is slightly more thermostable than the wild-type RC but not as stable as RC F(M197)H. Nevertheless, after the introduction of an additional H-bond of P with its protein environment, we succeeded in improving the resolution of the crystal structure up to 2.6 Å. For the first time, an analysis of the structural consequences of the I(L177)H mutation in the *C. sphaeroides* RC was performed based on its crystal structure obtained with 2.9 Å resolution (PDB ID: 3V3Z) [6]. The model of pigment-protein interactions that arose as a result of the mutation was somewhat refined in this work. It is shown that only one new water molecule, rather than two, appeared on the β-side of B_B_ and is suggested to be the 6^th^ –ligand of the Mg^2+^atom of this BChl. This water appears to donate H-bond to the imidazole group of His-L177, which is also in contact with the methyl group of the pyrrole ring-1 of BChl P_A_ (Figure 4). The distance between the N and C atoms of these groups in the structure is 2.06 Å, which exceeds known distance characteristic for C-N covalent bond (~1.5 Å) [25]. The resolution of the double mutant RC crystal structure is not yet sufficient to unequivocally confirm the covalent nature of BChl – protein binding and to reveal the origin of this bond and needs further improvement. However, it should be noted that the potential reactivity of this (B)Chl methyl group is supported by the fact that in the Chl *f* structure in the pyrrole ring-I, instead of a methyl group, a vinyl group is present [26].

The mutant RC I(L177)H + F(M197)H presented in this work turned out to be of interest as an object that has a modified H-bonding network near the BChl dimer P. Recently, the presence of H-bonding clusters on the periplasmic surface of the *C. sphaeroides* RC was reported [10]. Comparative analysis of such networks in the structures of natural and mutant RCs of different purple bacteria allowed us to assume that the polar protein environment can specifically affect the oxidative potential of the primary electron donor. In particular, it was suggested that when the π-electron system of P is connected with these networks, there is a correlation between the value of E_m_ P/P^+^ and the patterns, sizes, and, apparently, the total strength of H-bonds in the clusters. The crystal structure of RC I(L177)H + F(M197)H provides data in support of this assumption.

The main pattern of the H-bonding network in the protein environment of BChl P_B_ in the double mutant RC structure (Figure 4) was found to be similar to that observed in the F(M197)H mutant RC [10], with one exception. The 3.7 Å remoteness of the water-C from the side group of Asp-L155 exceeds the H-bond distance (Figure 6B). If the H-bond between Asp-L155 and water-C is not formed, one H-bond falls out of the H-bonding network. As a result, the bulk of the network connected with BChl P_B_ through imidazole His-M197 became not as extended as in the F(M197)H RC. We assume that this can explain why the value of E_m_ P/P^+^ in double mutant RC is ~20 mV lower than the sum of changes in the mid-point potentials caused by each of the two mutations. One should also not exclude the possibility that a more significant out-of-plane rotation of the acetyl group of P_B_ reduces its conjugation into the π-electron system of the macrocycle, affecting both the total strength of H-bonds in the network and the value of E_m_ P/P^+^. The loss of one H-bond in the network may also serve as an explanation for the fact that the thermal stability of RC I(L177)H + F(M197)H turned out to be less than that of RC F(M197)H [10]. A possible reason for the non-formation of the H-bond between Asp-L155 and water-C in RC I(L177)H + F(M197)H may be associated with the effect of the above-mentioned strong binding of BChl P_A_ with His L177, which is expected to shift P towards BChl B_B_. This assumption is consistent with more significant out-of-plane rotation of the C2-acetyl carbonyl group of P_B_, compared with that of RC F(M197)H.

H-bond networks on the periplasmic surface of purple bacterial RCs have long been noticed. The periplasmic surface of the *C. spheroids* RC is an area of short-term binding of soluble cytochrome c2, which donates electrons for oxidized P [27,28]. Acidic residues Asp-L155 and Asp-M184 (Figure 6 and Appendix A) were shown to be among a few others on the RC surface that interact with basic residues on cytochrome c_2_. In the recent work of Allen et al., it was proposed that the wide-spread network on the periplasmic side of RC participate in a hydrogen-bonded linear pathway that transfers protons between the environment of P^+^ and the membrane surface [29]. This pathway includes the amino acid residues Asp-L155, Ser-L158, Tyr-M198, Asn M195, and a few water molecules in the vicinity of P_B_ (Figure 6 and [10]). The possible role of the symmetry related cluster of H-bonds near BChl P_A_ remains to be clarified.

To summarize, the data obtained in this study clarify the details of the pigment-protein interactions that arose as a result of mutation I(L177)H. In addition, the crystal structure of RC I(L177)H + F(M197)H illustrates that protein surroundings can affect the oxidation potential of the primary electron donor by altering the parameters of the H-bonding networks connected with the π-electron system of P.

## Figures and Tables

**Figure 1 membranes-13-00157-f001:**
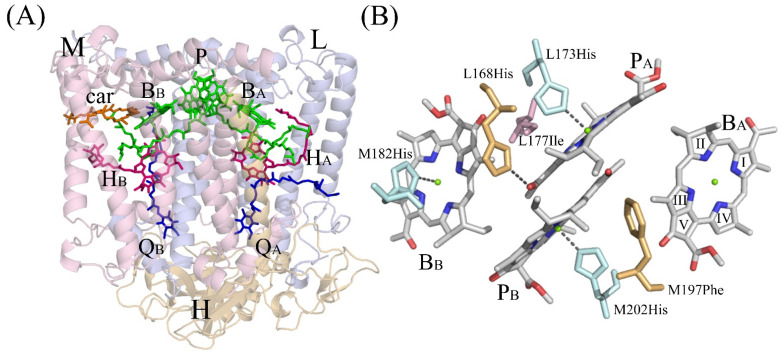
(**A**)–The overall structure of the reaction center of *C. sphaeroides* (PDB ID: 3v3y) and (**B**)-protein environment of bacteriochlorophylls. L, M, and H–correspondent RC subunits; P-BChl dimer of P_A_ and P_B_; B_A_ and B_B_-monomeric BChls; H_A_ and H_B_ - monomeric BPheos; Q_A_ and Q_B_-ubiquinones; car-carotenoid. Shown in pale-blue: His-M182, His-L173, and His-M202- ligands of BChls B_B_, P_A,_ and P_B_, correspondently. Symmetry related residues His-L168 and Phe-M197 are shown in pale-yellow, Ile-L177 is shown in light-pink. Pyrrole rings numbers are shown on the structure of BChl B_A_.

**Figure 2 membranes-13-00157-f002:**
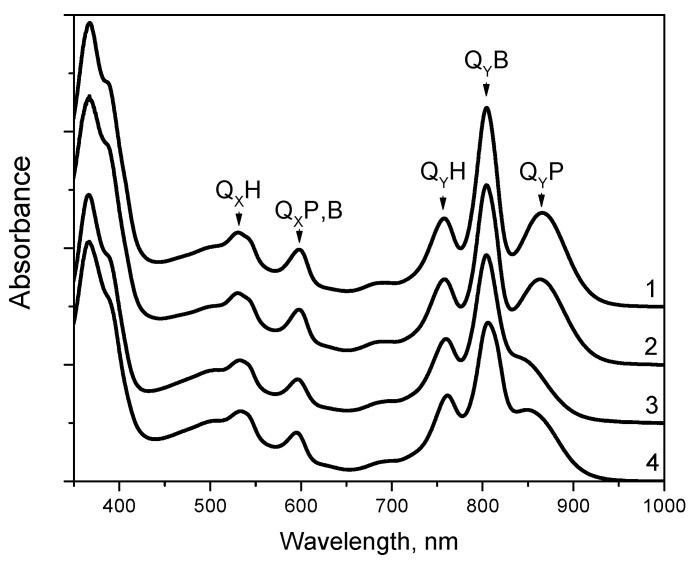
Electronic ground-state absorption spectra of *C. sphaeroides* reaction center from wild-type (1) and mutant strains F(M197)H (2), I(L177)H (3), and I(L177)H+F(M197)H (4). Spectra were measured at room temperature and normalized at Q_Y_H absorption band.

**Figure 3 membranes-13-00157-f003:**
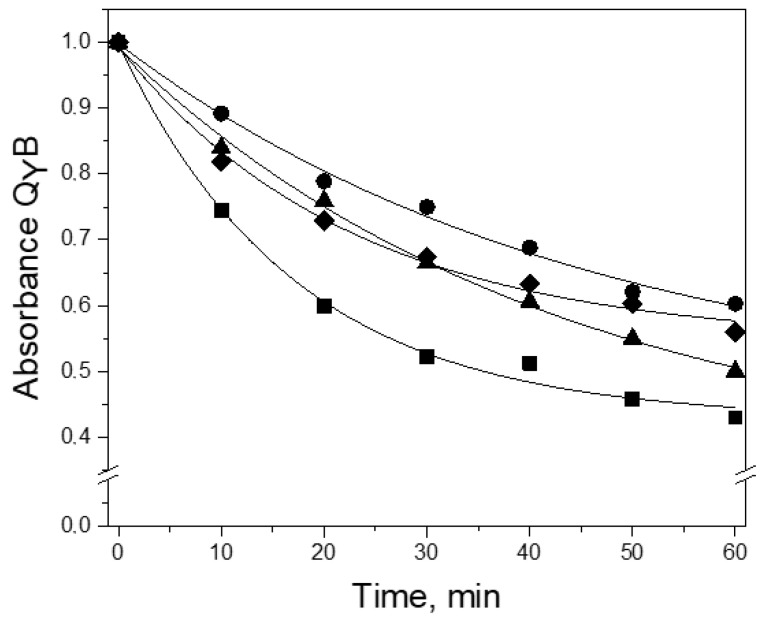
Changes in Q_Y_B absorption band amplitude in the spectra of the wild-type (▲), I(L177)H (■), F(M197)H (●), and I(L177)H+F(M197)H (♦) RCs during their incubation in Tris-Triton buffer at 48 °C.

**Figure 4 membranes-13-00157-f004:**
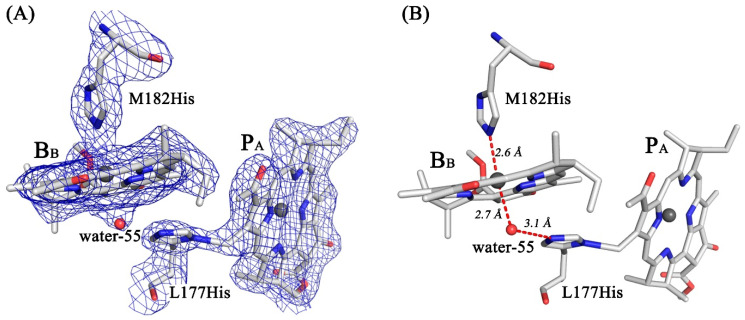
Electron density maps for RC I(L177)H + F(M197)H attributable to the P_A_ and B_B_ BChls as well as the His-L177 residue (**A**) and the structural model fitted to the density (**B**) (PDB ID: 8C3F).

**Figure 5 membranes-13-00157-f005:**
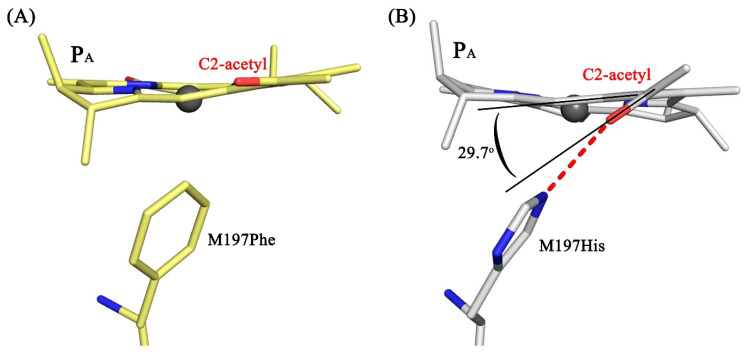
Position of the C2-acetyl group of BChl P_B_ in the structures of the wild-type RC (PDB ID: 3V3Y) (**A**) and RC I(L177)H + F(M197)H (PDB ID: 8C3F) (**B**). The view is along the BChl plane.

**Figure 6 membranes-13-00157-f006:**
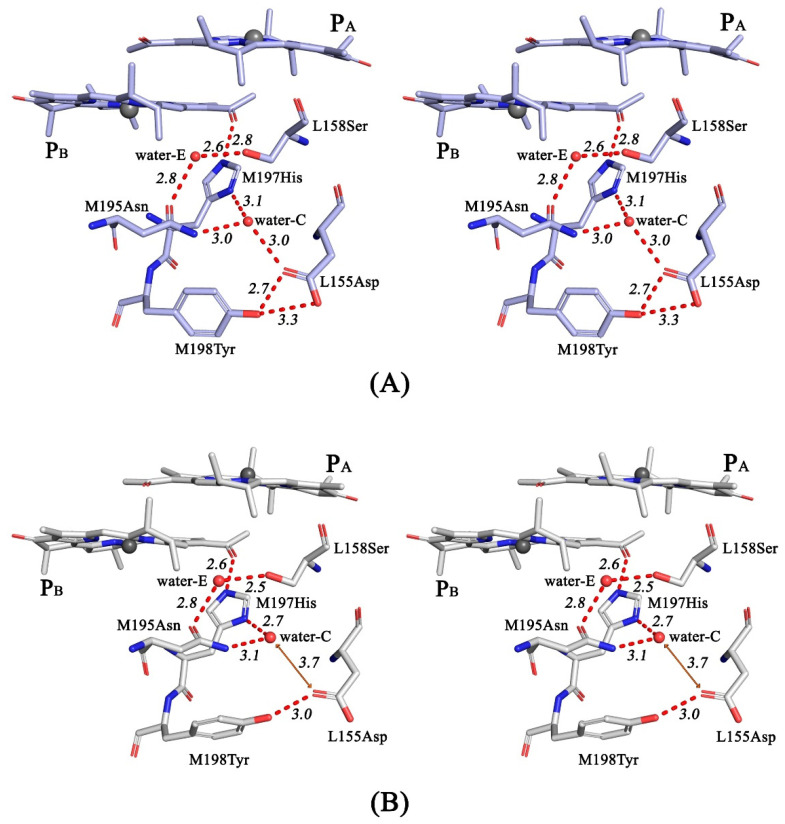
Stereoviews of the hydrogen-bonding networks in the proximity of the M197 residue in RC F(M197)H (**A**) and RC I(L177)H + F(M197)H (**B**). All distances are shown in Å.

**Table 1 membranes-13-00157-t001:** Data collection and processing.

Data Collection	
Diffraction source	PETRA III, beamline P11
Wavelength (Å)	1.03312
Temperature (K)	100
Detector	Pilatus 6M
Rotation range per image (°)	0.1
Total rotation range (°)	180
Space group	P3_1_21
*a*, *b*, *c* (Å)	139.3 139.3 184.6
Resolution range (Å)	50.00–2.60 (2.67–2.60)
Total No. of reflections	637 789 (43 719)
No. of unique reflections	64 208 (4 564)
Completeness (%)	99.7 (97.2)
Redundancy	9.9 (9.6)
〈 *I*/σ(*I*)〉	19.21 (1.04)
*R*_r.i.m._‡	10.7 (216.8)
CC_1/2_	99.9 (49.0)
Structure solution and refinement	
Resolution range (Å)	50.00–2.60 (2.63–2.60)
Completeness (%)	99.9 (100.0)
No. of reflections, working set	64 207 (3 995)
No. of reflections, test set	5 901 (200)
*R* _cryst_	19.43 (34.92)
*R* _free_	21.98 (38.88)
R.m.s. deviations	
Bonds (Å)	0.008
Angles (°)	1.15
Average *B* factors (Å^2^)	64.95
Protein	64.13
Ligand	72.15
Water	59.05
Calculated DPI (Å)	0.17
Maximal estimated error (Å)	0.31
Ramachandran plot	
Most favoured (%)	96.57
Allowed (%)	3.43

## Data Availability

MDPI Research Data Policies.

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
