# Peer review of "Properties and Crystal Structure of the Cereibacter sphaeroides Photosynthetic Reaction Center with Double Amino Acid Substitution I(L177)H + F(M197)H"

_membranes, 2023, doi:10.3390/membranes13020157_

Round 1

Reviewer 1 Report

This paper continues the extensive investigation of the bacterial RC by the authors. Having already studied the I(L177)H and F(M197)H mutants, they used the double mutant to (1) improve the crystal quality of the I(L177)H form and (2) examine the interactions between the two altered sites. The experiment was successful in improving crystal resolution, and was able to provide a plausible explanation for the observed spectrum and P/P+ midpoint potential for the double mutant.

The paper is well-written, with only a few minor typos and places where I feel word choice could be improved:

Line 244: a ) is missing

Line 331: a few characters are in Russian

Lines 57, 108, 357: suggest replacing "symmetrical" with "corresponding"

Line 61: suggest replacing "substitution of Ile-L177 for His" with "substitution of His for Ile-L177"

Line 217: suggest replacing "fits of the structural model to the density" with "structural model fitted to the density"

Line 249: suggest replacing "presents" with "is present"

Line 267: suggest replacing "remoteness" with "distance"

Author Response

Thank you for your valuable comments.

The minor typos and places were corrected according to your proposals:

Line 244: a ) is missing – corrected

Line 331: a few characters are in Russian – corrected

Lines 57, 108, 357: suggest replacing "symmetrical" with "corresponding"

We have changed "symmetrical" for “symmetry related”  since the cofactors and L and M protein subunits are related to each other by an approximate two-fold symmetry axis (Feher et al., 1989). We also added this sentence to the text of Introduction.

Line 61: suggest replacing "substitution of Ile-L177 for His" with "substitution of His for Ile-L177"

We introduce His instead of Ile- L177, so, to our opinion, "substitution of Ile-L177 for His" is correct.

Line 217: suggest replacing "fits of the structural model to the density" with "structural model fitted to the density"

Corrected according to your proposal

Line 249: suggest replacing "presents" with "is present"

Corrected according to your proposal

Line 267: suggest replacing "remoteness" with "distance"

Corrected according to your proposal

Reviewer 2 Report

Fufina et al. analyzed the spectroscopic properties of the photosynthetic reaction center with double amino acid substitution I(L177)H + F(M197)H from Cereibacter sphaeroides and interpreted their spectroscopic properties based on the crystal structure. The authors clearly clarified the estimated results of the photosyntheic reaction center in previous studies through this study. It is expected to contribute to the field of photosynthetic reaction center research. I support that the paper be published after a few corrections.

1. In the Experimental section, separate sections should be created for each experiment. The current version is very poorly readable. 

2. The figures of the crystal structure are of very low quality. The resolution needs to be improved.

3. line 47-60: "Two BChls PA and PB are combined into a dimer P that serves as a primary electron donor. An important property of P is the midpoint potential (Em) P/P+ which is related to the energy level of P* after light excitation. In the wild-type RCs of purple bacteria cofactors interact with integral L- and M-subunits through relatively weak noncovalent contacts. Figure 1-B shows four BChl molecules in the C. sphaeroides RC and some amino acid residues from their environment. Histidine residues, L173 and M202, coordinate central Mg2+ atoms of BChls PA and PB, respectively, and His-M182 is the ligand of BChl BB. In native C. sphaeroides RC the C2-acetyl carbonyl group of the BChl PA forms an H-bond with His-L168. The C2-acetyl carbonyl group of BChl PB is free of H-bonds with its environment since the residue closest to this group and symmetrical to His-L168 is Phe-M197. Crystal structures of photosynthetic RCs also contain a considerable amount of bound water molecules with few of them located in the hydrophobic core of the complexes (not shown, discussed below)" Authors should add reference sources relevant to their sentences.

Minor

-line 64: 'SDS PAGE' should be 'SDS-PAGE'

-line 233: "... BChl PB shows good structural conservation ..." remove "good"

-line 331: (Рис. 6-B). It is not English.

-line 355-356: "This pathway includes the amino acid residues Asp-L155, Ser-L158, Tyr-M198, Asn M195, and water molecules networked in the vicinity of PB (Fig. 6 and [7]."  The last part of the sentence needs to be corrected.

Author Response

Reviewer #2:

Thank you for your valuable comments.

  1. In the Experimental section, separate sections should be created for each experiment. The current version is very poorly readable. 

- We divided  Experimental section into few separate sections according to your recommendation.

  1. The figures of the crystal structure are of very low quality. The resolution needs to be improved.

- We have reloaded the figures with appropriate resolution

  1. line 47-60: "Two BChls PA and PB are combined into a dimer P that serves as a primary electron donor. An important property of P is the midpoint potential (Em) P/P+ which is related to the energy level of P* after light excitation. In the wild-type RCs of purple bacteria cofactors interact with integral L- and M-subunits through relatively weak noncovalent contacts. Figure 1-B shows four BChl molecules in the C. sphaeroides RC and some amino acid residues from their environment. Histidine residues, L173 and M202, coordinate central Mg2+ atoms of BChls PA and PB, respectively, and His-M182 is the ligand of BChl BB. In native C. sphaeroides RC the C2-acetyl carbonyl group of the BChl PA forms an H-bond with His-L168. The C2-acetyl carbonyl group of BChl PB is free of H-bonds with its environment since the residue closest to this group and symmetrical to His-L168 is Phe-M197. Crystal structures of photosynthetic RCs also contain a considerable amount of bound water molecules with few of them located in the hydrophobic core of the complexes (not shown, discussed below)" Authors should add reference sources relevant to their sentences.

We have added two more references in this part of the text according to your recommendation.

We have reloaded the figures with good resolution

Minor

-line 64: 'SDS PAGE' should be 'SDS-PAGE' - corrected

-line 233: "... BChl PB shows good structural conservation ..." remove "good" - corrected

-line 331: (Рис. 6-B). It is not English. - corrected

-line 355-356: "This pathway includes the amino acid residues Asp-L155, Ser-L158, Tyr-M198, Asn M195, and water molecules networked in the vicinity of PB (Fig. 6 and [7]."  The last part of the sentence needs to be corrected.

Corrected as following: “This pathway includes the amino acid residues Asp-L155, Ser-L158, Tyr-M198, Asn M195, and few water molecules in the vicinity of PB (Fig. 6 and [9]).

Sincerely yours,

Tatiana Fufna and co-authors.

Round 2

Reviewer 2 Report

The authors addressed the reviewer's concerns. The revised manuscript is suitable for publication.